# Genome-Wide Identification, Characterisation, and Evolution of the Transcription Factor WRKY in Grapevine (*Vitis vinifera*): New View and Update

**DOI:** 10.3390/ijms25116241

**Published:** 2024-06-05

**Authors:** Ekaterina Vodiasova, Anastasiya Sinchenko, Pavel Khvatkov, Sergey Dolgov

**Affiliations:** 1Federal State Funded Institution of Science “The Labor Red Banner Order Nikita Botanical Gardens—National Scientific Center of the RAS”, Nikita, 298648 Yalta, Russia; nas.sin4enko@gmail.com (A.S.); khvatkov1987@gmail.com (P.K.); dolgov@bibch.ru (S.D.); 2A.O. Kovalevsky Institute of Biology of the Southern Seas of RAS, 299011 Sevastopol, Russia; 3Branch of Shemyakin and Ovchinnikov Institute of Bioorganic Chemistry, 142290 Puschino, Russia

**Keywords:** WRKY transcription factor, grape, *Vitis vinifera*, genome-wide analyses, phylogeny, grape cultivars

## Abstract

WRKYs are a multigenic family of transcription factors that are plant-specific and involved in the regulation of plant development and various stress response processes. However, the evolution of *WRKY* genes is not fully understood. This family has also been incompletely studied in grapevine, and *WRKY* genes have been named with different numbers in different studies, leading to great confusion. In this work, 62 *Vitis vinifera WRKY* genes were identified based on six genomes of different cultivars. All *WRKY* genes were numbered according to their chromosomal location, and a complete revision of the numbering was performed. Amino acid variability between different cultivars was assessed for the first time and was greater than 5% for some WRKYs. According to the gene structure, all *WRKYs* could be divided into two groups: more exons/long length and fewer exons/short length. For the first time, some chimeric *WRKY* genes were found in grapevine, which may play a specific role in the regulation of different processes: VvWRKY17 (an N-terminal signal peptide region followed by a non-cytoplasmic domain) and VvWRKY61 (Frigida-like domain). Five phylogenetic clades A–E were revealed and correlated with the WRKY groups (I, II, III). The evolution of *WRKY* was studied, and we proposed a *WRKY* evolution model where there were two dynamic phases of complexity and simplification in the evolution of *WRKY*.

## 1. Introduction

Living organisms are constantly exposed to various negative factors. Plants have an attached way of life and have been forced to develop special mechanisms to cope with the harmful effects of the environment. One of the ways to increase resistance is through the evolution of an intricate signalling system that allows them to respond rapidly to external stimuli. The key molecules of signalling systems are transcription factors (TFs) that determine the response of plants to biotic and abiotic stress [1,2,3].

One of the largest multigene families of transcription regulators in higher plants is the WRKY TFs [4]. These proteins are important regulators that are involved in the response to various biotic and abiotic factors [5,6,7,8,9,10,11,12,13]. They are integral components of numerous facets of the plant innate immune system and play pivotal roles in plant growth and development [14,15,16].

The WRKY transcription factors possess a DNA-binding domain (DBD) at the N-terminal and a zinc-finger motif at the C-terminal of the protein [17]. The sixty amino acid-long domain is characterised by the signature motif ‘WRKYGQK’. WRKY proteins are divided into four groups according to the number of WRKY domains and zinc finger pattern [18]. The WRKYs belonging to group I have two WRKY DBDs and a Cys2-His2-type zinc finger motif; group II contains proteins with a single DBD and a Cys2-His2-type zinc finger motif; group III possesses proteins with a single DBD and a Cys2-His/Cys-type zinc finger motif; and group IV contains proteins with an incomplete WRKY domain without a zinc finger [19,20]. Concurrently, some *WRKY* genes with three domains (*Glycyrrhiza uralensis*, *Lupinus angustifolius*, *Gossypium raimondii*, *Linum usitatissimum*) and even with four (*Aquilegia coerulea*, *Oryza nivara*, *Solanum lycopersicum*) have been identified [16]. Although the WRKY DNA-binding domain is highly conserved, there is increasing evidence of amino acid substitutions in the ‘WRKYGQK’ motif [14]. Such variability has been described in maize [21], banana [22], soybean [23], and rice [24].

To date, WRKY transcription factors have been identified in more than 150 plants, according to the PlantTFDB database. The number of genes encoding this TF varies significantly among plants. The lowest number (<20) of *WRKY* genes was found for algae or other poorly studied plants (6 *WRKY* genes in *Helianthus annuus*) (yellow column in Figure 1B). The limited number of *WRKY* genes described for some plant species is likely due to the lack of genomic data. For example, only 46 genes were identified for *B. napus* in 2009 [25]. However, following the availability of the first genomic data in 2016, the number of described genes increased to 287 [26]. Another study identified 278 genes [27], while the PlantTFDB database described 285 genes.

A few plants have a relatively small number of *WRKYs*, including *Amborella trichopoda* (32), *Coffea canephora* (49), *Genlisea aurea* (38), *Carica papaya* (49), and *Lactuca sativa* (50), as reported in PlantTFDB. The maximum number of *WRKY* genes was identified in *Brassica napus* (285), *Glycine max* (296), and *Panicum virgatum* (275). The distribution of the number of *WRKY* genes in different plant species, as reported by PlantTFDB, is represented in Figure 1. The majority of plants have between 50 and 130 genes encoding WRKY. Even closely related species exhibit differences in the number of genes. For example, *Brassica napus* has 285 genes, *Brassica oleracea* has 191 genes, and *Brassica napa* has 180 genes.

The considerable diversity in the number of genes present in different plants serves to illustrate the complex evolution of the *WRKY* gene family. As previously stated, these TFs are involved in the regulation of a multitude of biological processes, including the response to stressful biotic and abiotic factors. Therefore, the large number of duplication events may be an adaptation mechanism to unfavourable external stimuli.

At present, many studies have been carried out on the effect of various stresses on WRKYs expression in different plants [14,15,16,28,29,30,31,32,33]. The principal stress factors under investigation are presented in Figure 1. As WRKYs play a pivotal role in plant defence in response to abiotic stress, a genetic engineering approach could be an effective strategy for enhancing tolerance to some negative factors [34,35,36]. The grapevine (*Vitis vinifera* L.) is one of the most widely cultivated plant species for commercial purposes. Global production of this crop reaches 70 million tonnes, occupying more than seven million hectares of land for harvesting [37]. In 2016, the value of grapes in agricultural enterprises reached $68 billion, making it the most valuable fruit crop in the world. *Vitis vinifera* is mainly used to produce a variety of commodities, including wine, table grapes, sultanas, grape juice concentrate, and spirits for industrial use [38]. Adaptation to climate change is an important step in the future of viticulture, which depends heavily on weather and climatic conditions [39]. The primary challenge identified is the heightened risk of water stress, which may result in reduced yield in terms of both quantity and quality, given that a water deficit is observed in the majority of grape-producing regions [40,41,42,43]. Consequently, it is of paramount importance to study the functioning of this transcription factor family in grapevine and to identify *WRKY* genes that respond to one or another type of biotic and abiotic factor.

To date, several genome-wide analyses of *WRKY* genes in grapevine have been performed. The number of genes encoding *WRKY* found varied and appeared to depend on the version of the genomic assembly analysed. A number of papers have identified 59 genes based on the 12X assembly of the *V. vinifera* cv. Pinot Noir (PN40024) genome sequences and the NCBI GenBank database [37,44,45]. The PlantTFDB transcription factor database also lists 59 *WRKY* for *V. vinifera*, the same number of genes analysed in other studies [7,46]. When analysing the NCBI GenBank database, 80 *WRKY* genes were predicted, which is probably due to the presence of heterozygosity in loci [47]. With the appearance of a more complete genomic assembly of grape (NCBI accession number GCA_030704535) and an improved annotation of the first genomic assembly (GCA_000003745, v3 annotation), a revision of the *WRKY* genes was conducted, resulting in the identification of 61 genes [48,49]. Nevertheless, all research on the identification of this transcription factor has been conducted on a single Pinot Noir cultivar (PN40024). Consequently, the chromosomal position of some *WRKYs* remains unconfirmed, and the total number of these genes, including heterozygous variants, remains unknown.

Also, the functional role of different *VvWRKY* genes was investigated in several studies where the authors in transgenics of *A. taliana* or *N. tabacum* revealed the effect of *VvWRKY* genes by overexpression in plant tissues [50,51,52,53,54,55,56,57,58,59,60,61,62,63]. The function of WRKY transcription factors has also been studied in transgenic grapes [64,65,66,67,68,69]. In these studies, the WRKY gene numbers were determined by the level of homology to Arabidopsis WRKY or by homology with the already known RefSeq or PlantTFDB databases. This has resulted in confusion and a lack of a unified systematics of grape WRKY, as previously mentioned [70]. Consequently, it is difficult to utilise the results of these studies.

This research is devoted to the identification of WRKY using genome-wide analyses based on several grape cultivar assemblies (including diploid). Phylogenetic analyses, protein structure, and intra-varietal variability were employed to perform a comprehensive revision of grape WRKY proteins and propose a novel classification.

## 2. Results

### 2.1. The Phylogeny of Grape WRKY Genes

The domain analysis found WRKY DNA-binding domain proteins in each grapevine assembly. However, the number of proteins identified differed between grape cultivars. A total of 234 proteins were found in the diploid assembly for Cabernet Franc, 181 for Cabernet Sauvignon, and 304 for Pinot Noir clone FPS123. In the haploid assemblies of Pinot Noir clone PN40024, the number of proteins identified was also variable and depended on the annotation. These were 89 for the 12X assembly (GCA_000003745.2), 87 for the RefSeq reference assembly (GCA_030704535), 65 for the GenBank reference assembly (GCA_030704535), and 84 for the assembly from GrapeGenomics database. This discrepancy in the number of proteins can be attributed to the presence of multiple variant isoforms for a single gene, the heterozygosity of loci, and the annotation pipeline. Given that some WRKY proteins identified in the 12X assembly and the RefSeq reference assembly have identical IDs, the two datasets were merged into a single entity, and only unique IDs, of which 97 were found, were retained for further analysis. Thus, the combined set of WRKY amino acid sequences from six grape genomic assemblies contained 965 proteins, including all isoforms.

A phylogenetic analysis was conducted to reveal the phylogenetic relationships among all 965 complete amino acid sequences. The analysis identified 62 clusters with 100 bootstrap supports, which were found to be significantly different from neighbouring clusters (Figure 2). 

Each cluster corresponds to a specific VvWRKY TF and contains a variable number of amino acid sequences from different grape cultivars. The phylogenetic tree with expanded clusters is presented in Appendix A. For each gene encoding the corresponding WRKY protein, its position on the chromosome was determined according to each grape cultivar genome. All gene IDs and chromosome locations are given in Appendix A. WRKY TFs were numbered according to the ordinal chromosome location number (in most of the assemblies analysed).

The exon-intron structure and conserved motifs were analysed for 62 WRKY proteins of the reference grape genome for Pinot Noir cl. PN40024 v. 5 from the GrapeGenomics Database (Figure 2). The number of exons ranged from 2 up to 8. Twenty-five motifs were found for 62 protein sequences from *V. vinifera* cultivar Pinot Noir cl. PN40024 v. 5 (GrapeGenomics database). All motif sequences in grape WRKY proteins are represented in Table 1. All WRKYs except for VvWRKY26 were revealed to have three conserved motifs arranged one after the other: 1-3-2. Motif 1 is characterised by the presence of the WRKY signature. Motifs 4 and 13 also contain the WRKY signature. Based on the analysis of the number of DNA-binding domains and the zinc finger motif (Appendix A), all studied proteins were assigned to a specific WRKY type for grape described before (groups I, IIa–IIe, III) [37,44,45,47].

A phylogenetic analysis revealed clustering between the different WRKY classes. Five evolutionary groups (A–E) were identified based on the complete amino acid sequences of the proteins. The WRKY groups (I, IIa–IIe, III) were completely correlated with the evolutionary groups. Within some groups, subclades with 100 bootstrap supports were distinguished.

Group A is associated with WRKY type III. It does not contain subclusters, exhibits a similar exon-intron structure (three exons), and has a similar set of conserved motifs. Motifs 11 and 15 in the N-terminal region of the protein are distinctive characteristics of this group.

Group B (the group of WRKY IIc) is comprised of four subclades, each with 100 bootstrap supports (B1–B4). Subgroup B1 is distinguished by a motif 16 in the C-terminal region, which is unique to it and exhibits a three-exon structure. Subgroup B4 is characterised by a simple structure and a two-exon configuration.

Group C is distinguished by genetic diversity, differences in the composition of conserved motifs in different WRKYs, complex structure (the number of exons varies from 4 to 8), and long length. WRKYs of this phylogenetic group belong to the WRKY type I category, as they contain two conserved domains containing the WRKY signature. The C-terminal WRKY domain contains conserved motif 1, while the N-terminal WRKY domain varies in different proteins and is characterised by motif 4 (in VvWRKY11, VvWRKY37, VvWRKY29, VvWRKY42, VvWRKY27, VvWRKY18, VvWRKY15, VvWRKY3) or motif 13 (in VvWRKY61, VvWRKY33, VvWRKY62, VvWRKY49). No clustering into subgroups is observed in C.

In group D, as in group C, *WRKY* genes exhibit a distinct composition of conserved domains and a complex exon-intron structure (the number of exons varies from 3 to 6). This evolutionary group corresponds to groups IIa + IIb. Three clusters with 100 supports are distinguished, each lacking characteristic conserved motifs or exon-intron structure (D1–D3). The D3 subgroup corresponds to the WRKY IIa type. Group D is distinguished by the presence of two characteristic motifs, 7 and 8, which are located on the N-terminal and C-terminal, respectively. Subgroups D1 and D2 (IIb) are characterised by the presence of a conserved motif 10, which follows the WRKY signature.

Group E is distinguished by a gene structure comprising three exons and the presence of two subgroups, which corresponds to the division into two types, IId and IIe. Subgroup E1 (IId) exhibits characteristic motifs 14 and 18 on the N-terminal and motif 9, followed by motif 1 with tetra amino acid residues (WRKY). Subgroup E2 (IIe, respectively) has the simplest set of conserved motifs of all WRKYs (only common for all 1-3-2 motifs).

There are genes that differ from the others (Figure 2, black arrows). VvWRKY17 and VvWRKY30 represent distinct evolutionary branches and cannot be assigned to either phylogenetic group. VvWRKY44 does not possess motifs 11 and 15, which are characteristic of the entire group A. VvWRKY26 lacks motif 1, which contains tetra-amino acid residues, but has zinc fingers.

### 2.2. The Characteristics of 62 WRKY Classes in Grapes

Phylogenetic analysis enabled the classification of all WRKYs in different grape cultivars and assemblies (Table 2). For each protein, the coding gene, its corresponding locus on the chromosome, and the number of genes encoding each VvWRKY class (Ng) were identified. The number of isoforms (Ni) varied for a single gene from 1 to 36.

In some cases, a large number of isoforms were determined, even in haploid assemblies. For *VvWRKY1*, *VvWRKY10*, *VvWRKY15*, *VvWRKY22*, *VvWRKY29*, *VvWRKY33*, *VvWRKY49*, and *VvWRKY62*, the number of isoforms exceeded 10 in the diploid assembly and exceeded 5 in the haploid one. The analysis of haploid assemblies revealed that, with the exception of *VvWRKY42*, each transcription factor class had a single locus in the genome. For *VvWRKY42*, three loci were identified in the genome of Pinot Noir clone 40024 (v.5, Grape Genomics), two of which were located on chromosome 12 and one on chromosome 9. Upon analysis of diploid-phased assemblies of three grape cultivars, it was observed that two genes encoding a particular transcription factor were most often identified. This is consistent with the diploid-phased assembly. However, instances were noted where only one locus in the genome or three (potentially indicative of duplication) were detected.

Analyses of the chromosomal location of *WRKY* genes revealed that they are absent from the third chromosome. There are regions of chromosomes where there are clusters of several genes, but there are also chromosomes with only one or two *WRKY* genes. *WRKY* genes of the same class in different grape cultivars are located on the same chromosome, with the exception of the genes encoding the transcription factors VvWRKY42 and VvWRKY56. The *VvWRKY42* gene in the Pinot Noir cl. PN40024 is located on chromosomes 9 and 12, while the *VvWRKY56* gene in the Pinot Noir cl. FPS123 is located on chromosomes 2 and 16 (Figure 3).

It was observed that some WRKYs were not present in all grape cultivars. For instance, *VvWRKY26* and *VvWRKY53* were not found in Cabernet Franc cl. 04, while *VvWRKY4* and *VvWRKY26* were not found in Cabernet Sauvignon cl. 08. Furthermore, differences were also identified between the various assemblies of the reference cultivar Pinot Noir cl. PN40024, likely due to differences in the assembly and annotation pipelines employed. A comprehensive list of all analysed and classified WRKY proteins, along with their respective IDs and chromosomal locations, is provided in Appendix A.

For each WRKY, the amino acid variability within the class, mutations in DNA-binding hepta-peptide sequences, and the type of zinc finger motif were analysed. Only the 62 amino acid WRKY sequences from *V. vinifera* cultivar Pinot Noir cl. PN40024, v. 5 (Grape Genomics database) (see Table 2 for gene IDs) were retained to analyse length, position on the chromosomes, number of exons, functional family determination (FF:number), WRKY domain location, and presence of other domains in the proteins, as this is the only assembly in which all WRKYs were detected. The results are presented in Table 3.

The length of VvWRKY ranged from 151 to 746 aa and the number of exons from 2 to 8. The mean value of genetic distances between varieties ranged from 0 (VvWRKY12) to 0.113 (VvWRKY17). Variability of the conserved DNA-binding heptapeptide WRKYGQK was detected. VvWRKY8, VvWRKY13, VvWRKY14, and VvWRKY24 possess the WRKYGKK amino acid sequence, whereas in VvWRKY17, a mutation Arg to Lys has occurred in the characteristic tetrapeptide WRKY. The WRKY domains were analysed, and functional families were identified according to the CATH databases.

The analysis of complete protein sequences rather than domain regions alone, enabled the identification of chimeric forms among VvWRKY transcription factors that include other domains in addition to DBD WRKY. The following chimeric VvWRKYs were identified in grapes: a Zn-cluster domain (IPR018872) followed by a WRKY domain (1); a Frigida-like domain upstream of two WRKY domains (IPRO12474) (2); the structure motif COILS (3), which is a coil located upstream or downstream of the domain; and the signal peptide in the N-terminal and noncytoplasmic domain (VvWRKY17) (4). The LxLxLx repressor and the LxxLL co-activator motifs were identified in eight and twelve VvWRKY proteins, respectively (Table 3).

A phylogenetic analysis of 62 amino acid sequences of VvWRKY domains from *V. vinifera* cultivar Pinot Noir clone PN40024, v. 5 (Grape Genomics database) revealed a similar tree topology as for the complete protein sequences of this gene family (Figure 4).

Clade B (WRKY group IIc), clade C (I), and clade D (IIa + IIb) are sister lineages and form a single large cluster. This is also observed in the tree based on the complete protein sequences (Figure 2). At the same time, the two trees exhibit a difference in topologies. Clade E (groups WRKY IIe + IId) and clade A (III) constitute a single clade with high bootstrap support, whereas this was not observed when the complete protein sequences were examined.

An association between the clustering of WRKY groups with the functional family of the protein, WRKY signature sequences, and the presence of other domains in the protein with the formation of chimeric WRKY TFs was revealed. Thus, clade D (IIa + IIb) is characterised by the presence of the structural motif COILS Coil. This motif is also present in VvWRKY16 (IIc), VvWRKY21 (IIc), and VvWRKY54 (IIe). The Zn-cluster domain characterises the subclade group E2 (IId). The heptapeptide sequence WRKYGKK is present only in subclade B2 (group IIc).

An analysis of the *WRKY* gene structure revealed a bipartite distribution of both protein lengths (approximately 300 and 500 amino acids) and exon numbers (three and five, respectively) (Figure 5A,C). This indicates the presence of *VvWRKY* genes with varying degrees of complexity in their structure. Analyses of exon number and protein length in relation to the evolutionary clades of *VvWRKY* genes demonstrated that clades C and D exhibited a more complex structure, while the other *VvWRKYs* exhibited a more simplified gene structure (Figure 5B,C).

### 2.3. Unified Systematics of Grape WRKY Genes

Each *WRKY* gene was matched with IDs from the most common databases: PlantTFDB, NCBI (RefSeq, GenBank), Ensembel, Uniprot, and the previously used names of each WRKY transcription factor (Table 4). A homology analysis of all amino acid sequences revealed a correspondence between the new numbering proposed by us (based on the localisation of genes on chromosomes) and all other protein IDs and WRKY numbers used previously. The confusion in gene names is evident from the table. Most of the genes have more than three different names, and some of them have as many as six different names. For example, the gene *VvWRKY31* had the names *VvWRKY14*, *VvWRKY30*, *VvWRKY40*, *VvWRKY28*, and *VvWRKY4* in different databases and studies. It was also found that the RefSeq database (which is also included in the reference genome annotation) has genes that have the same numbers but belong to different VvWRKY classes. For example, VvWRKY5, VvWRKY53, and VvWRKY54 are described in the RefSeq database as VvWRKY22 (XP_010658402, XP_002276925, and XP_010662789).

## 3. Discussion

The plant-specific transcription factor WRKY is involved in plant development and in the response to various biotic and abiotic stresses. This gene family has been studied in many species, including the model plant *Arabidopsis* and important crops such as rice, cucumber, coffee, tomato, etc. [70,71,72,73]. Despite the large number of studies on grapes, the *WRKY* genes have not been fully studied in this crop. To date, there are several important studies that have focused on the search for *WRKY* genes in grapes [37,44,45,47]. These studies were based on analyses of the genome of only one cultivar, *V. vinifera* cv. Pinot Noir (PN40024) (GCA_000003745), where the assembly has been continuously improved and re-annotated. This has led to a variation in the number of genes identified (from 59 to 80), their length, the number of exons, and the clarification of the different motifs present in the above studies.

To date, two chromosome-level assemblies are available at NCBI for Pinot Noir, which is accepted as a reference. The assembly GCA_000003745, known as 12X, is the first assembly of the grape genome and already has assembly version 3 [74]. This assembly has already been analysed in previous studies. In August 2023, a new assembly, GCA_030704535, appeared, which became the reference assembly for this species and was not previously analysed for WRKY studies [75]. The Grape Genomics database also contains the *V. vinifera* cultivar Pinot Noir cl. PN40024, v. 5, with its annotation [75,76]. In our work, 62 *WRKY* genes were identified based on the analysis of all three assemblies mentioned above. The length range of the found VvWRKYs is from 151 to 746 aa and differs from the previously identified ones from 101 to 612 aa [37]. The number of exons ranges from 2 to 8, while the previously identified ranges were 1–17 [45] and 2–7 [37]. The differences in length and exon-intron structure are explained by the incomplete assembly of GCA_000003745 compared to today.

### 3.1. Grapevine WRKY Numbering

Since there is no unified justification for naming genes in this family, a great deal of confusion has arisen over the years of studying WRKY in grapes. First, some researchers named the gene according to the number of the homologous *WRKY* gene in a model plant, mainly *Arabidopsis* [10,50,54,56,57,58,59,60,61,62,63,64,66,77,78,79,80,81]. It seems unreasonable to use homology to another species in this multigene family for numbering, since the number of genes encoding WRKY TFs varies greatly in different plants. The second group of researchers numbered the genes according to their position on the chromosome [37,44,45,70]. Initially, because not all genes had a clear localisation, unknown chromosomes were left in the assemblies, and much confusion arose. A recent study on *WRKY* gene classification also failed to conclusively determine the position of *WRKY59* (an unknown chromosome) [70].

At the same time, a uniform numbering system is necessary because confusion is already emerging. For example, in work on transgenic grapes, it was shown that WRKY70 (VIT_13s0067g03140) is involved in norisoprenoid and flavonol biosynthesis [69]. At the same time, the name WRKY70 is found in the RefSeq database under two IDs: XP_00227272504 (in our study VvWRKY28) and XP_002275401 (in our study VvWRKY45). Thus, naming VvWWRKY70 could lead to errors in further studies. The situation with the VvWRKY35 gene is different. Its role in tolerance to cold and salt stress in transgenic Arabidopsis was previously studied, where it was described as VvWRKY28 [55]. In another paper investigating the functions of WRKYs, this gene (GSVIVT01021397001) is referred to as VvWRKY22 [7], and in a paper investigating WRKYs in strawberries, it is included in the phylogenetic analysis as VvWRKY71 (XP_002272089) [82]. At the same time, another VvWRKY71 with accession number XP_002283603 is deposited in the RefSeq database and analysed under this name in another paper on grape [83]. A recent paper was published showing that VvWRKY71 could promote the biosynthesis of proanthocyanidins, but it is extremely difficult to understand which RefSeq ID was used to annotate the assembled transcriptome [84]. Such situations show the extreme necessity of using a unified *WRKY* gene numbering system.

Our genome-wide analysis, based on several complete genome assemblies, has determined the chromosomal position for all 62 genes found. Therefore, we propose to name the genes according to their location on the chromosome in the reference genome of *V. vinifera* cv. Pinot Noir cl. PN40024, v.5 [75,76] with an annotation dated September 2023 (GenBank annotation). For the convenience of researchers, not only was a complete revision of the numbering carried out, but correspondences were also found between the proposed new numbering and all the most common IDs from different databases and previous *WRKY* gene numbers (Table 4).

### 3.2. Grape WRKY Diversity in Cultivars

With the development of NGS technologies and the accumulation of a large amount of data, there are studies on the comparison of several genomes within a species, which allows us to assess the presence of intraspecific nucleotide variability in genes. Thus, on the basis of comparative analyses of GATA transcription factors among 19 Arabidopsis genomes, intraspecific amino acid variability was shown [85]. Sequence diversity of the WRKY transcription factor family has been demonstrated in wild and cultivated barley, where the authors showed that haplotype and nucleotide diversity in the majority of *WRKY* genes were higher in the wild barley population [86]. The high nucleotide variability could lead to further misestimation of relative expression by RT-PCR if primers hit no conservative region of the gene. This situation is probably observed in studies of WRKY expression in response to pathogenic fungal infection and SA treatment. As the authors identified only 57% of the differentially expressed genes (in contrast to 70% of the DEGs in Arabidopsis), they suggested that the reason for this difference was poor primer annealing in the studied cultivar, due to the primer design using the other cultivar [47].

In our study, *WRKY* transcription factor genes were analysed for the first time in different grape varieties (Cabernet Franc, Cabernet Sauvignon, and Pinot Noir clones FPS123 and PN40024). For some *WRKY* genes, inter-varietal amino acid variability was detected (Table 3), which was greater than 5% for the genes *VvWRKY3*, *VvWRKY4*, *VvWRKY17*, *VvWRKY24*, *VvWRKY35*, *VvWRKY42*, and *VvWRKY54*. It should also be noted that some *WRKY* genes are not present in all varieties. For some genes, there was a significant difference in the number of isoforms in different cultivars (Table 2). For example, the transcription factor VvWRKY1 is encoded by a single gene, but the number of predicted isoforms differed: 12 isoforms for Cabernet Franc, 10 for Cabernet Sauvignon, 15 for Pinot Noir cl. FPS123, but only 4 for Pinot Noir cl. PN40024. This seems to be explained by the fact that for Pinot Noir cl. PN40024, the assembly is haploid, and for the other cultivars, the assembly is diploid. However, VvWRKY33 has 10 isoforms for Pinot Noir cl. FPS123 and only 2 isoforms each for Cabernet Franc and Cabernet Sauvignon. It is obvious that the variability of some WRKYs is quite high, which may indirectly influence, for example, the resistance of different varieties to stress factors. This issue needs further study.

### 3.3. WRKY Domains in Grapevine

WRKY transcription factors are characterised by a specific gene structure: they contain a DNA-binding domain with a conserved WRKY motif (WRKYGQK) in the L-terminal and a zinc finger motif in the C-terminal. Trp, Tyr, and two Lys residues in the heptapeptide sequence are known to be essential for DNA binding [87,88,89]. However, there is known diversity in the WRKY signature sequence [16,37,45,90]. Similar to previous studies, we detected grape transcription factors with WRKYGKK (VvWRKY8, VvWRKY13, VvWRKY14, VvWRKY24) [37,45,47]. Variants of WRKY domains with such hepta amino acid residues are consistently found in other plants: rice, OsWRKY7 [91], tobacco, NtWRKY12 [92], pepper, CaWRKY39, CaWRKY50, CaWRKY56, and CaWRKY62 [93], cucumber CsWRKY41, CsWRKY44, and CsWRKY54 [94]. Thus, this sequence is likely to be plant-specific, but the efficiency of binding to WK- and W-boxes is not fully understood [91].

Also in grapevine, the *VvWRKY17* gene was found to contain the conservative tetrapeptide sequence WKKY instead of the traditional WRKY, which is confirmed by other studies [37,45]. This variant is frequently found in different species, indicating that this mutation is fixed [28,90]. It is likely that genes with this motif have some functionality, and such mutations could result from altered conditions and be needed for the development of plant resistance [16,95,96].

We first discovered *VvWRKY26*, which contains a WRKY domain but has a previously unknown WRKY signature sequence, WMKGNPH. Whether this gene is functional is unknown and requires a separate study. The zinc-finger motif in the C-terminal also has some variability. The majority of VvWRKYs have C_2_H_2_, except for six VvWRKYs that have a C_2_HC zinc-finger motif, according to us and previous studies [37,44,45,47]. VvWRKY26 has an altered C2HY motif, which also casts doubt on the functionality of this gene.

### 3.4. WRKY Groups and Evolution Clades

Based on the number of WRKY domains and the type of zinc finger motif, there is an accepted classification of this family of transcription factors, as discussed in the introduction (I–IV). Seven major groups and subgroups have been identified in flowering plants: I, II (IIa, IIb, IIc, IId, IIe), and III [19]. Our results complement previous studies in grapes and show that 12 genes of WRKY group I, 43 genes of WRKY group II, 6 genes of WRKY group III, and *VvWRKY26*, which probably belongs to group IV because it lacks the C_2_H_2_ or C_2_HC zinc finger motif, are present in grapes. WRKY group II is the most numerous and, according to previous studies, is divided into several subgroups IIa–IIe. We have also shown the presence of two genes, *VvWRKY17* and *VvWRKY30*, which do not belong to the known subgroups but are clearly assigned to group II.

Subtypes IIa–IIe were distinguished as a result of clustering of phylogenetic trees, which led to confusion between the concepts of “WRKY group based on number of domains and type of zinc finger motif” and “WRKY group based on evolution”. This led to confusion. For example, in a previous study, two *VvWRKY* genes were characterised as NG (non-group), although these genes had one WRKY domain and a C_2_H_2_ zinc-finger motif and should have been assigned to group II [37]. This situation arose because it was previously thought that the division into groups I–III reflected the evolution of this family of transcription factors. With the emergence of new data, it became clear that this was not the case. Recent papers investigating the evolution of WRKYs in plants repeatedly state that group II cannot be considered an evolutionary group. If it were a phylogenetically distinct group, then all subgroups (IIa-IIe) should be part of it, but this is not observed: groups IIa + IIb, IIc, and IIe + IId are phylogenetically distinct groups [16,90]. Furthermore, it has been shown that divergence within major evolutionary lineages occurs differently in dicots and monocots, and that the division into subgroups IIa–IIe is not universal for all plants [90].

Therefore, to avoid confusion, in our study we propose to introduce an evolutionary classification—phylogenetic clades A–E, which are distinguished as a result of analyses of the complete amino acid sequences of WRKY TFs (Figure 4) (for ease of comparison, we indicate which WRKY group corresponds to a particular clade). In our work, we have shown that clades B and D have a more complex evolution: there is a divergence into subclades B1–B4 and D1–D3. These subclades have different conserved motifs that may play a role in protein functionality. However, the topology based on phylogenetic analysis of domain regions alone does not reflect the true evolution (Figure 2 and Figure 4). For example, subclade B2 is present on both trees and differs from other genes by a modified WRKY motif (WRKYGKK instead of WRKYGQK). At the same time, subclade B1 is formed only when the whole protein is analysed, as it is characterised by a conserved motif 16 in the C-terminal region. The topology also differs at the clade level. Thus, in domain analysis, clades C and D form a branch, whereas in complete sequence analysis, clade C clusters with B, as confirmed by studies in 30 plant species, from green algae to *Arabidopsis* [97]. These findings suggest the need to analyse complete amino acid sequences, not just domain regions.

### 3.5. Chimeric WRKY

We also discovered some new chimeric WRKY TFs in grapes. These are genes that do not contain only the WRKY domain. Such proteins could be involved in the regulation of several seemingly disparate processes and have previously been found in different plant species [16,17,90,98,99]. There are 370 known domain architectures in the InterPro database for the WRKY domain (IPR003657) (as of 30 April 2024). There are 22,847 proteins with a single WRKY domain architecture and 5670 with two WRKY domains. The most common chimeric WRKY TF variant is the Zn cluster domain (IPR018872), followed by the WRKY domain (2975 proteins). In grapes, the Zn cluster domain chimeric protein is characteristic of subgroup IId (subclade E1), which is in agreement with other studies [16]. However, its functional role is not known. Domains such as disease resistance domain, kinase domain, or leucine-rich repeat, which are commonly found in plants, were not found in grapes. However, a structural coil motif was found that is characteristic of clade D. It is likely that these proteins are WRKYs of the immune signalling pathway.

The new chimeric proteins are VvWRKY17 and VvWRKY61. VvWRKY17 has an N-terminal signal peptide region followed by a non-cytoplasmic domain (according to the Phobius prediction). This WRKY transcription factor is likely to have a specific localisation. The localisation of WRKY in different cell compartments has been shown in *Glycyrrhiza glabra* [11]. This issue requires further investigation.

VvWRKY61 has a Frigida-like domain (IPR012474), followed by two WRKY domains. This family represents proteins similar to the FRIGIDA protein. This protein is located in the nucleus and is required for the regulation of flowering time [100]. AtWRKY75 in *Arabidopsis* has previously been shown to be involved in the regulation of flowering [101]. Chimeric WRKYs containing the ZF_SBP domain associated with flower development have also been found, and it has been suggested that such chimeric proteins may play an important role in flowering [90]. Thus, a VvWRKY61 TF with a Frigida-like domain and two WRKY domains may regulate flowering in grapes, which requires further investigation.

### 3.6. Grapevine WRKY Evolution

Our phylogenetic analysis of grapevine *WRKY* suggests the most likely evolutionary process for this family of transcription factors. There are many theories for the evolution of *WRKY* genes from the unicellular early green lineage to multicellular plants [11,18,19,24,91,97]. The ancestor of all *WRKY* genes is thought to be an N-terminal addition of a WRKY-like motif to the BED finger-like C_2_HC zinc finger domain [97]. Subsequently, gene duplication occurred in the charophyte green algae, and genes containing two WRKY domains were formed and found in the unicellular green alga *Chlamydomonas reinhardtii* [18,19,70]. More recently, the presence of a gene with a single domain belonging to clade D (IIb) was shown in the filamentous terrestrial alga *Klebsormidium flaccidum* [102]. Thus, clades C (group I) and D (IIb) are early lineages. Phylogenetic clades A (III), E (IIe and IId), and B (IIc) evolved from clade C (I) due to the loss of the NTWD [91,97]. However, the origin of clade D (IIb and IIa) remains unknown. There are two theories according to which clade D (IIb + IIa) evolved from a single-domain ancestor (IIa + IIb separate hypothesis) or from clade C (I) (Group I hypothesis) [97]. More recently, it has been shown that clade D is characterised by the presence of a V-type intron, whereas clades A, B, and E possess a conserved R-type intron present in CTWDs [91]. As a result, the authors lean towards the “IIa + IIb separate hypothesis”.

Our studies also confirm that the most likely evolutionary pathway was the origin of clade D from a common ancestor. Gene duplication is known to be one of the major evolutionary mechanisms generating new diversity and often leading to gene paralogy [103,104]. In grapevine, approximately 40% of *WRKY*s originate from tandem or segmental duplications [37]. Duplication events lead to an increase in the number of members of evolutionary clades and reflect an ongoing evolutionary process. There are other evolutionary processes, such as the gain or loss of introns. An intron has a complex structure (donor and acceptor splice sites, branch point, polypyrimidine tract, and appropriate splicing enhancers) and can be several thousand nucleotides long. Therefore, the emergence of new introns by the gradual accumulation of functional sub-elements is unlikely, and it has been shown that introns can also arise by segmental genomic duplication [105]. At the same time, intron loss has been shown to be more likely than intron gain [106], making intron gain rarer. In our study, we showed that clades D and C differ significantly in intron number and protein length from other evolutionary groups (Figure 5). If clade D had evolved from clade C as a result of the loss of NTWD, there would most likely have been a loss of some introns and a reduction in the length of the transcription factor. This is what we observe in genes from clades A, B, and E that evolved from clade C. And we observe the opposite situation: clade D is equivalent to clade C in protein length and number of introns. The *WRKY* genes from clade D have a large, conserved motif 7 and a coil in the N-terminal structural motif. Considering that the intron gain process is quite rare, the probability that clade D lost NTWD initially and then gained a new domain is quite low.

Based on the probability of evolutionary events, we suggest that there were two dynamic phases of complexity and simplification in the evolution of *WRKY* (Figure 6). The two clades C and D evolved from a common ancestor as a result of the complexity of the gene structure, then divergence occurred, and a characteristic motif (NTWD or Coil) emerged in each clade. This theory is supported by the fact that subclade D3 (IIa) evolved from clade D (IIb), which is also characterised by a reduction in length and number of introns (Table 3). The proposed evolutionary model is consistent with the fact that macroevolutionary patterns are characterised by the periodicity of opposing dynamics. There is a general pattern of evolution consisting of two distinct evolutionary phases: a short, explosive innovation phase leading to a dramatic increase in genome complexity, and a longer simplification phase leading to either loss of genetic material or adaptive ordering of the genome [106,107,108,109]. Quantitatively, genome evolution is dominated by reduction and simplification, followed by episodes of increasing complexity, which is also consistent with the proposed theory.

## 4. Materials and Methods

### 4.1. Genome Data

Two databases representing genomic assemblies of grapes were analysed: the National Center for Biotechnology Information (NCBI) and GrapeGenomics (https://grapegenomics.com/ accessed on 24 February 2024). The NCBI currently contains 19 genomic assemblies of varying quality for different grape cultivars: three assemblies have a chromosomal level, ten have a scaffold level, and six have a contig level. For this study, we analysed the genome assembly of *V. vinifera* cultivar Pinot Noir cl. PN40024 (GCA_030704535, August 2023), which is the reference for the grape [75], and the first assembly of *V. vinifera* cultivar Pinot Noir cl. PN40024 is known as 12X (GCA_000003745.2) [74]. The assembly GCA_030704535 has two annotations (RefSeq and GenBank), which are included in our analyses. No predicted proteins are represented for other chromosomal-level assemblies, and thus they were excluded from the analyses. Furthermore, lower-level assemblies (scaffolds or contigs) were not included in the analysis.

Twenty-three genomic assemblies for different cultivars were available in the second database, Grape Genomics. Three assemblies with diploid-phased chromosomal levels and predicted protein data were selected from this database: *V. vinifera* cultivar Cabernet Franc cl. 04, *V. vinifera* cultivar Cabernet Sauvignon cl. 08, and *V. vinifera* cultivar Pinot Noir cl. FPS123 [110]. Also, the reference assembly of *V. vinifera* cultivar Pinot Noir cl. PN40024, v. 5 was represented in this database [75,76]. Only those assemblies that were at the chromosome level and had protein predictions were included in the analysis. Thus, six genomic assemblies and seven protein annotations were included in the analyses.

### 4.2. Genome-Wide Analyses and Identification of WRKY

The genome-wide analyses were performed through six grape genome assemblies. For each genomic assembly, all predicted protein sequences and CDS data were downloaded and further analysed. The identification of WRKY proteins was based on finding the WRKY domain signature using InterProScan software, version 5.63-95.0 [111] and the Pfam protein family database [112]. After that, all proteins that had the WRKY DNA-binding domain (accession number PF03106 according to the Pfam database) were checked for containing other domains, and functional families (FF) were detected using the Pfam, InterPro, and CATH databases [113,114]. Zinc finger, LXXLL, and LXLXLX motifs were identified manually.

The exon-intron structure of each WRKY transcription factor-encoding gene was detected using the CDS description and genome annotation of accordance grapevine cultivars from the NCBI and GrapeGenomics databases. A motif analysis was performed using the MEME online tool in classic mode (https://meme-suite.org, accessed on 15 April 2024). The location information of the grape WRKY genes was obtained from the Genome Data Viewer available in NCBI and GrapeGenomics for each assembly. The amino acid diversity analyses of WRKY genes were conducted using MEGA software (version 11.0.13) [115,116]. The average evolutionary divergence over different grape cultivars for each WRKY gene was estimated as the number of amino acid substitutions per site from averaging over all sequence pairs with a bootstrap procedure (1000 replicates). The analysis was conducted using the Poisson correction model [117].

To search for accordance among WRKY proteins from PlantTFDB, NCBI, Uniprot, and Ensembl databases, we used similarity and homology analyses between protein sequences performed using local BLAST v. 2.9.0 with a threshold E-value of 1 × 10^−10^ [118,119,120,121,122,123].

### 4.3. Phylogenetic Analyses

A phylogenetic analysis was conducted using IQ-TREE software, version 2.3.0 [124]. The dataset comprised all identified WRKY protein sequences of *V. vinifera* from six genome assemblies. The multiple alignment of protein sequences was performed using the MAFFT online service (version 7.0) with the scoring matrix BLOSUM62 [125,126]. The best amino acid substitution model, according to BIC, was estimated with ModelFinder [127]. The consistency of the Maximum Likelihood (ML) tree was validated by an ultrafast bootstrap value of 1000 [128]. The final phylogenetic tree was visualised with FigTree version 1.4.4. The rooting of trees was according to the midpoint.

## 5. Conclusions

Sixty-two *WRKY* transcription factor genes have been identified in *Vitis vinifera* grapes. The structure of each s*VvWRKY* gene was studied, and its chromosomal location was determined. A new numbering of the genes according to their chromosomal location in the reference genome of *V. vinifera* cv. Pinot Noir cl. PN40024, v.5 was suggested. Inter-varietal amino acid variability was revealed, reaching 5% for some genes. Chimeric *VvWRKY* genes were also found, which may have specific regulatory functions. Phylogenetic analysis shows that the evolution of *WRKY* genes went through a phase of complication (intron gain and formation of genes with two domains) and a phase of simplification (loss of introns and reduction of protein length). The data obtained indicate a high functional flexibility of this family, which is consistent with the wide range of processes in which WRKY transcription factors are involved.

## Figures and Tables

**Figure 1 ijms-25-06241-f001:**
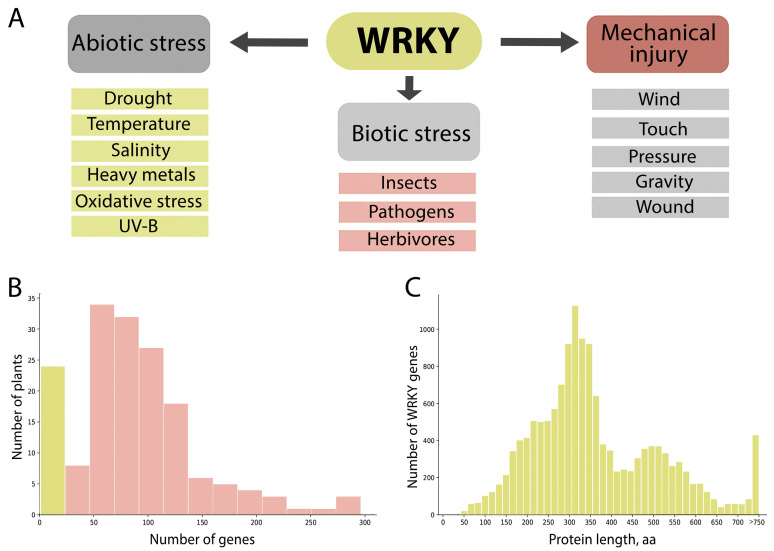
The characterisation of the WRKY family of plant transcription factors. The investigation of the influence of abiotic and biotic factors on WRKY TF expression (**A**). The distribution of *WRKY* gene number in different plant species (**B**) and *WRKY* protein length distribution (**C**) based on PlantTFDB database data.

**Figure 2 ijms-25-06241-f002:**
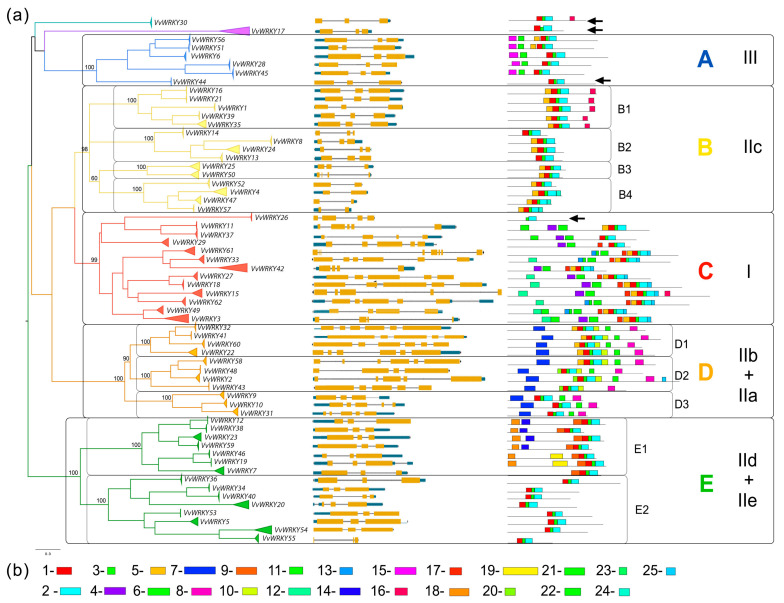
The evolutionary relationship, exon-intron structure, and motif analysis of WRKY transcription factor family members in *V. vinifera* (**a**). The midpoint-rooted phylogenetic tree was constructed with the Maximum Likelihood (ML) method and the VT + F + R4 substitution model (on the left) based on the complete amino acid sequences. The different colours and letters (A–E) note the five phylogenetic groups; the letters with numbers indicate the subgroups. The accepted type of WRKY classification is indicated on the right (I, IIa–IIe, III). The yellow boxes represent exons, and lines represent introns (in the middle). All exon and intron lengths are drawn to scale. The 23 different coloured boxes represent diverse conserved motifs identified with MEME (on the right). The numbering of conservative motifs is noted below (**b**).

**Figure 3 ijms-25-06241-f003:**
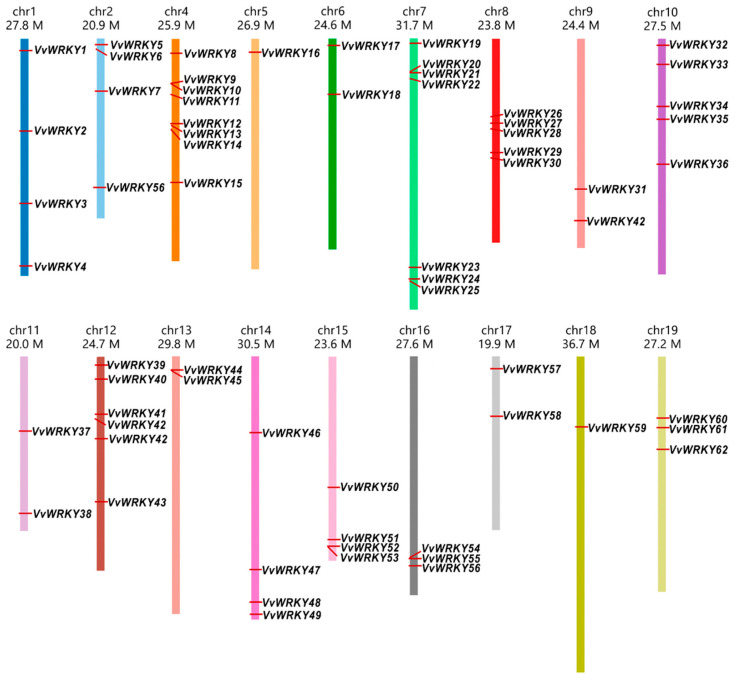
Chromosomal location of 62 genes of transcriptional factor WRKY on the nineteen grape chromosomes.

**Figure 4 ijms-25-06241-f004:**
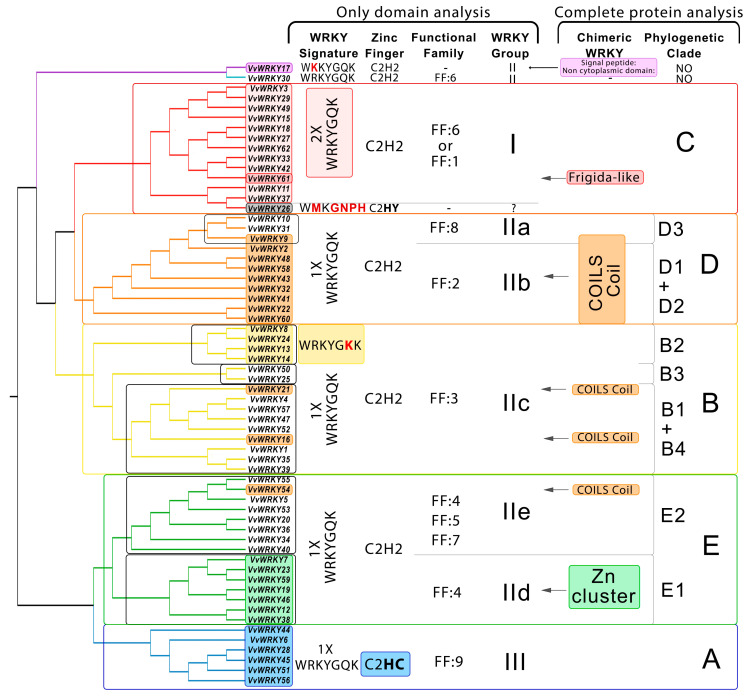
The topology of the phylogenetic tree associated with domain characteristics. The midpoint-rooted phylogenetic tree was constructed with the Maximum Likelihood (ML) method and the Q.plant + G4 substitution model based on the domain amino acid sequences. The different colours and letters (A–E) indicate the five phylogenetic groups according to Figure 2. The coloured boxes note the features of different groups of *WRKY* genes.

**Figure 5 ijms-25-06241-f005:**
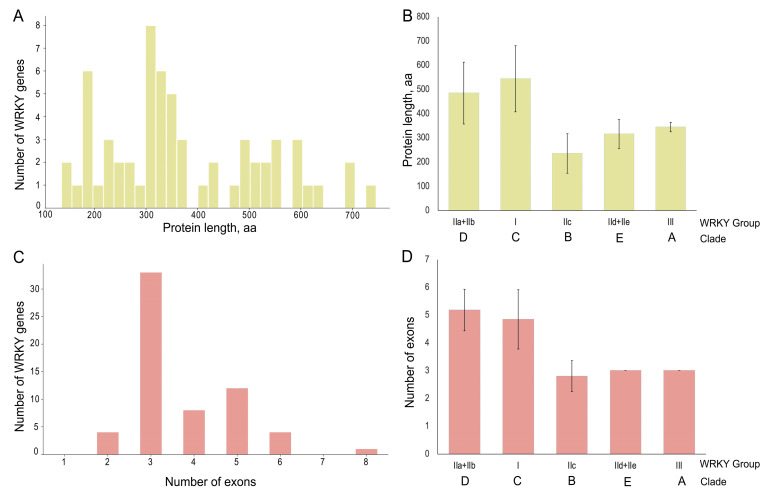
The analysis of length and gene structure. The distribution of VvWRKY protein length (**A**) and exon number (**C**) in grapevine. The mean values of protein length (**B**) and exon number (**D**) according to belonging to different phylogenetic clades and WRKY groups. The data are presented as mean ± SD.

**Figure 6 ijms-25-06241-f006:**
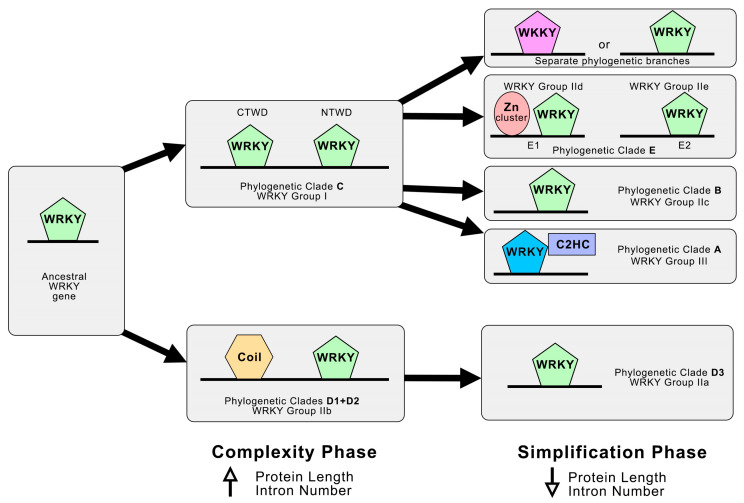
The suggested evolutionary model of the WRKY transcriptional factor family according to the phylogenetic studies in grapevine.

**Table 1 ijms-25-06241-t001:** The amino acid consensus sequences of conserved motifs in grapevine WRKYs.

Motif	Motif Consensus Amino Acid Sequences	Width, aa
1	DILDDGYRWRKYGQKPVKGSP	21
2	GCPVRKQVZRSSEDPSIVITTYEGKHNHP	29
3	YPRSYYRCTSA	11
4	DGYNWRKYGQKQVKGSEYPRSYYKCTYPNC	30
5	KKKAZKTIREPRVAVQTRSEV	21
6	ERSHDGQITEIIYKGTHNHPKPQPNRRSALG	31
7	KDELEVLKAELERVREENEKLREMLEQITKBYNALQMHLVEJMQ	44
8	TVEAATAAITADPNFTAALAAAITSIIG	28
9	SSGRCHCSKRRKLRVKRSIRVPAISNKIA	29
10	LPPAATAMASTTSAAASMLLS	21
11	MASISASAPFPTITLDLTQ	19
12	PSPLPIARSPYFTIPPGLSPTSLLDSPVLLS	33
13	EDGYNWRKYGQKQVKGSE	18
14	DCREIADYAVSKFKKVISJLNRTGHGRFR	29
15	MEEKLSWEQKTLINELTQGRELAKQLKIHL	33
16	LLRDHGLLQDIVPSFIRK	18
17	DEDDEDEPESKRRKKEV	18
18	QEAIQEAASAGLESVEKLIRLLSHAQDQ	28
19	QQMKHQADMMYRRSNSGINLKFDGSSCTPTMSSTRSFISSLSMDGSVANL	50
20	PVKKARVSVRARCDT	15
21	QGPFGMSHQZVLAQVTAQAAQAQSHMQLQ	29
22	REDLVVKILRSFEKALSILKCGG	23
23	NPRSYYKCTNA	11
24	EKPTDNFEHILNQMQ	15
25	VPAARNSSHBTAG	13

**Table 2 ijms-25-06241-t002:** The distribution of WRKY protein isoforms (Ni) and encoding genes (Ng) in different grape cultivars and assemblies.

Cultivar	Cabernet Franc		Cabernet Sauvignon		Pinot Noir								
Clone	04		08		FPS123		PN40024						
Assembly	Diploid						Haploid, 12X, and Reference				Haploid, Assembly v. 5		
	Grape Genomics						RefSeq		GenBank		Grape Genomics		
Gene Name	Ni	Ng	Ni	Ng	Ni	Ng	Ni	Ng	Ni	Ng	Ni	Ng	Gene ID
*VvWRKY1*	12	2	10	2	15	2	1	1	1	1	4	1	Vitvi01g04492.t001
*VvWRKY2*	4	2	1	1	2	2	2	1	1	1	3	1	Vitvi01g04490.t001
*VvWRKY3*	2	1	3	2	6	3	1	1	1	1	1	1	Vitvi01g02157.t001
*VvWRKY4*	1	1	-	-	2	2	1	1	1	1	2	1	Vitvi01g01680.t002
*VvWRKY5*	4	2	6	3	2	2	2	1	1	1	1	1	Vitvi02g00039.t001
*VvWRKY6*	2	2	2	2	2	2	1	1	2	1	2	1	Vitvi02g00114.t001
*VvWRKY7*	3	2	2	2	2	1	1	1	1	1	1	1	Vitvi02g01847.t001
*VvWRKY8*	2	2	2	2	2	2	1	1	-	-	1	1	Vitvi04g00133.t001
*VvWRKY9*	4	2	2	2	6	2	1	1	1	1	3	1	Vitvi04g04524.t001
*VvWRKY10*	14	2	14	2	12	2	2	1	1	1	1	1	Vitvi04g00511.t001
*VvWRKY11*	2	2	2	2	4	2	1	1	1	1	3	1	Vitvi04g04525.t001
*VvWRKY12*	4	2	2	1	4	2	1	1	1	1	1	1	Vitvi04g00756.t001
*VvWRKY13*	6	2	1	1	2	2	2	1	1	1	2	1	Vitvi04g00760.t001
*VvWRKY14*	2	2	1	1	2	2	1	1	1	1	1	1	Vitvi04g01985.t001
*VvWRKY15*	3	2	2	2	36	2	5	1	2	1	1	1	Vitvi04g01163.t001
*VvWRKY16*	2	2	2	2	2	2	1	1	1	1	1	1	Vitvi05g00145.t001
*VvWRKY17*	2	2	3	2	4	2	2	1	1	1	1	1	Vitvi06g01574.t001
*VvWRKY18*	7	2	4	2	2	2	1	1	1	1	1	1	Vitvi06g00741.t001
*VvWRKY19*	8	2	4	1	6	2	3	1	1	1	1	1	Vitvi07g00026.t001
*VvWRKY20*	6	2	4	2	6	2	3	1	1	1	2	1	Vitvi07g00421.t001
*VvWRKY21*	2	2	2	2	2	2	1	1	1	1	1	1	Vitvi07g00434.t001
*VvWRKY22*	12	2	2	2	12	2	2	1	1	1	1	1	Vitvi07g00523.t002
*VvWRKY23*	4	2	4	2	4	2	1	1	1	1	1	1	Vitvi07g01694.t001
*VvWRKY24*	2	2	2	2	4	2	1	1	1	1	2	1	Vitvi07g04782.t002
*VvWRKY25*	2	2	2	2	2	2	2	1	1	1	1	1	Vitvi07g01860.t001
*VvWRKY26*	-	-	-	-	2	2	-	-	1	1	1	1	Vitvi08g04106.t001
*VvWRKY27*	6	2	3	1	6	2	1	1	1	1	1	1	Vitvi08g00793.t001
*VvWRKY28*	2	2	1	1	4	2	1	1	1	1	1	1	Vitvi08g00868.t001
*VvWRKY29*	9	2	5	1	5	2	7	1	1	1	2	1	Vitvi08g01134.t001
*VvWRKY30*	2	2	1	1	2	2	1	1	1	1	1	1	Vitvi08g01221.t001
*VvWRKY31*	1	1	9	3	8	2	1	1	1	1	1	1	Vitvi09g01122.t001
*VvWRKY32*	6	2	4	2	4	2	1	1	1	1	1	1	Vitvi10g00063.t001
*VvWRKY33*	2	2	2	2	10	2	4	1	1	1	1	1	Vitvi10g00270.t001
*VvWRKY34*	2	1	4	2	4	2	1	1	1	1	1	1	Vitvi10g00618.t001
*VvWRKY35*	4	2	2	1	4	2	1	1	1	1	1	1	Vitvi10g00732.t001
*VvWRKY36*	2	2	2	2	3	3	1	1	1	1	1	1	Vitvi10g01078.t001
*VvWRKY37*	6	2	4	2	5	2	3	1	1	1	1	1	Vitvi11g00694.t002
*VvWRKY38*	4	2	4	2	4	2	1	1	1	1	1	1	Vitvi11g01188.t001
*VvWRKY39*	2	2	2	2	3	3	1	1	1	1	1	1	Vitvi12g00048.t001
*VvWRKY40*	2	2	2	2	2	2	1	1	1	1	1	1	Vitvi12g00148.t001
*VvWRKY41*	2	2	1	1	4	2	1	1	1	1	1	1	Vitvi12g00388.t001
*VvWRKY42*	1	1	2	2	5	3	4	1	3	3	3	3	Vitvi12g00664.t003 Vitvi12g04520.t001 Vitvi12g04129.t001
*VvWRKY43*	4	2	2	2	3	3	1	1	2	1	1	1	Vitvi12g01676.t001
*VvWRKY44*	2	2	1	1	2	2	1	1	1	1	1	1	Vitvi13g00189.t001
*VvWRKY45*	2	2	1	1	2	2	1	1	1	1	1	1	Vitvi13g01916.t001
*VvWRKY46*	4	2	4	2	4	2	1	1	1	1	1	1	Vitvi14g00540.t001
*VvWRKY47*	3	2	2	2	2	2	1	1	1	1	1	1	Vitvi14g01523.t001
*VvWRKY48*	2	2	1	1	2	2	3	1	1	1	1	1	Vitvi14g01907.t001
*VvWRKY49*	1	1	2	1	10	2	1	1	1	1	2	1	Vitvi14g02007.t001
*VvWRKY50*	4	2	4	2	6	2	1	1	1	1	1	1	Vitvi15g00539.t001
*VvWRKY51*	4	2	4	2	4	2	1	1	1	1	1	1	Vitvi15g01003.t001
*VvWRKY52*	1	1	2	2	2	2	1	1	1	1	1	1	Vitvi15g01087.t001
*VvWRKY53*	-	-	2	2	2	2	1	1	1	1	1	1	Vitvi15g01090.t001
*VvWRKY54*	3	3	2	2	3	3	1	1	1	1	1	1	Vitvi16g01132.t001
*VvWRKY55*	1	1	2	2	2	2	1	1	1	1	1	1	Vitvi16g01133.t001
*VvWRKY56*	2	2	2	2	2	2	1	1	1	1	1	1	Vitvi16g01213.t001
*VvWRKY57*	3	3	2	2	2	2	1	1	1	1	1	1	Vitvi17g00102.t001
*VvWRKY58*	3	2	6	3	4	2	2	1	1	1	1	1	Vitvi1700556.t001
*VvWRKY59*	6	2	3	1	5	2	1	1	1	1	1	1	Vitvi18g00742.t001
*VvWRKY60*	4	2	1	1	6	2	2	1	1	1	1	1	Vitvi19g00530.t001
*VvWRKY61*	4	2	4	2	1	1	1	1	1	1	1	1	Vitvi19g00617.t001
*VvWRKY62*	16	3	7	2	22	3	4	1	-	-	5	1	Vitvi19g04652.t002
Total	234	115	181	106	304	129	97	61	65	62	84	64	

**Table 3 ijms-25-06241-t003:** Characterisation of identified grapevine VvWRKY transcriptional factors.

Gene Name	*ORF, aa*	Exons	Mean Distance	WRKY Domain Location	DNA-Binding Residues	Zinc Finger	Functional Family	WRKY Group/Clade	Other Features
*VvWRKY1*	305	3	0.001	162–218	WRKYGQK	C2H2	FF:3	IIc/B	-
*VvWRKY2*	594	5	0.012	245–303	WRKYGQK	C2H2	FF:2	IIb/D	COILS Coil: 60–101 LxLxLx motif
*VvWRKY3*	502	4	0.051	247–303 427–484	WRKYGQK	C2H2	FF:6 FF:1	I/C	-
*VvWRKY4*	189	2	0.064	111–168	WRKYGQK	C2H2	FF:3	IIc/B	-
*VvWRKY5*	330	3	0.030	144–200	WRKYGQK	C2HC	FF:7	IIe/E	-
*VvWRKY6*	342	3	0.005	133–192	WRKYGQK	C2HC	FF:9	III/A	-
*VvWRKY7*	323	3	0.042	254–310	WRKYGQK	C2H2	FF:4	IId/E	Zn-cluster: 204–250 LxxLL motif
*VvWRKY8*	166	3	0.005	104–161	WRKYGKK	C2H2	FF:3	IIc/B	-
*VvWRKY9*	258	4	0.021	97–155	WRKYGQK	C2H2	FF:8	IIa/D	COILS Coil: 14–34 LxLxLx motif
*VvWRKY10*	317	5	0.010	160–218	WRKYGQK	C2H2	FF:8	IIa/D	LxxLL motif LxLxLx motif
*VvWRKY11*	491	5	0.004	139–196 354–410	WRKYGQK	C2H2	- FF:6	I/C	-
*VvWRKY12*	338	3	0.000	261–317	WRKYGQK	C2H2	FF:4	IId/E	Zn-cluster: 210–257
*VvWRKY13*	191	3	0.008	103–160	WRKYGKK	C2H2	FF:3	IIc/B	-
*VvWRKY14*	136	3	0.008	53–111	WRKYGKK	C2H2	FF:3	IIc/B	-
*VvWRKY15*	700	5	0.044	234–290 452–509	WRKYGQK	C2H2	FF:6 FF:6	I/C	LxLxLx motif
*VvWRKY16*	309	3	0.001	155–212	WRKYGQK	C2H2	FF:3	IIc/B	COILS Coil: 107–127 LxxLL motif
*VvWRKY17*	189	3	0.113	92–148	WKKYGQK	C2H2	-	NG	Signal peptide: 1–21 Non cytoplasmic domain: 22–189 LxLxLx motif
*VvWRKY18*	603	5	0.004	256–312 427–484	WRKYGQK	C2H2	FF:6 FF:1	I/C	-
*VvWRKY19*	340	3	0.001	274–331	WRKYGQK	C2H2	FF:4	IId/E	Zn-cluster: 226–270
*VvWRKY20*	242	3	0.030	48–105	WKKYGQK	C2H2	FF:4	IIe/E	-
*VvWRKY21*	302	3	0.003	149–206	WRKYGQK	C2H2	FF:3	IIc/B	COILS Coil: 102–122
*VvWRKY22*	512	6	0.041	269–326	WRKYGQK	C2H2	FF:2	IIb/D	COILS Coil: 105–132
*VvWRKY23*	336	3	0.045	265–321	WRKYGQK	C2H2	FF:4	IId/E	Zn-cluster: 215–261 LxxLL motif
*VvWRKY24*	193	3	0.068	106–163	WRKYGKK	C2H2	FF:3	IIc/B	-
*VvWRKY25*	226	3	0.026	150–206	WRKYGQK	C2H2	FF:3	IIc/B	-
*VvWRKY26*	236	4	0.005	66–109	WMKGNPH	C2HY	-	I/C	-
*VvWRKY27*	552	5	0.026	230–286 393–450	WRKYGQK	C2H2	FF:6 FF:1	I/C	-
*VvWRKY28*	334	3	0.008	136–196	WRKYGQK	C2HC	-	III/A	LxLxLx motif
*VvWRKY29*	477	5	0.029	196–250 397–453	WRKYGQK	C2H2	FF:6 FF:6	I/C	-
*VvWRKY30*	299	3	0.006	112–168	WRKYGQK	C2H2	FF:6	NG	-
*VvWRKY31*	311	5	0.020	160–218	WRKYGQK	C2H2	FF:8	IIa/D	-
*VvWRKY32*	535	6	0.004	277–334	WRKYGQK	C2H2	FF:2	IIb/D	COILS Coil: 99–133
*VvWRKY33*	626	5	0.019	270–326 437–493	WRKYGQK	C2H2	-	I/C	-
*VvWRKY34*	278	3	0.003	78–135	WRKYGQK	C2H2	FF:4	IIe/E	-
*VvWRKY35*	438	3	0.056	181–238	WRKYGQK	C2H2	FF:3	IIc/B	-
*VvWRKY36*	438	3	0.006	220–277	WRKYGQK	C2H2	FF:5	IIe/E	-
*VvWRKY37*	500	4	0.004	190–245 363–419	WRKYGQK	C2H2	- FF:6	I/C	-
*VvWRKY38*	297	3	0.006	226–282	WRKYGQK	C2H2	FF:4	IId/E	Zn-cluster: 175–222 LxxLL motif LxLxLx motif
*VvWRKY39*	311	3	0.004	173–230	WRKYGQK	C2H2	FF:3	IIc/B	-
*VvWRKY40*	244	3	0.008	76–133	WRKYGQK	C2H2	FF:4	IIe/E	-
*VvWRKY41*	593	5	0.005	311–368	WRKYGQK	C2H2	FF:2	IIb/D	COILS Coil: 130–171
*VvWRKY42*	407	4	0.069	110–166 285–342	WRKYGQK	C2H2	FF:6 FF:3	I/C	-
*VvWRKY43*	487	5	0.005	232–290	WRKYGQK	C2H2	FF:2	IIb/D	COILS Coil: 81–101 LxLxLx motif
*VvWRKY44*	364	3	0.001	175–235	WRKYGQK	C2HC	-	III/A	-
*VvWRKY45*	313	3	0.003	111–170	WRKYGQK	C2HC	-	III/A	LxxLL motif
*VvWRKY46*	365	3	0.004	298–354	WRKYGQK	C2H2	FF:4	IId/E	Zn-cluster: 249–294
*VvWRKY47*	182	2	0.034	105–162	WRKYGQK	C2H2	FF:3	IIc/B	-
*VvWRKY48*	555	4	0.001	225–283	WRKYGQK	C2H2	FF:2	IIb/D	COILS Coil: 33–81 LxLxLx motif
*VvWRKY49*	529	4	0.046	233–289 415–472	WRKYGQK	C2H2	FF:6 FF:1	I/C	-
*VvWRKY50*	228	4	0.033	155–211	WRKYGQK	C2H2	FF:3	IIc/B	-
*VvWRKY51*	349	3	0.004	119–178	WRKYGQK	C2HC	FF:9	III/A	-
*VvWRKY52*	201	2	0.006	123–180	WRKYGQK	C2H2	FF:3	IIc/B	LxxLL motif
*VvWRKY53*	348	3	0.001	168–225	WRKYGQK	C2H2	FF:7	IIe/E	-
*VvWRKY54*	329	3	0.058	159–215	WRKYGQK	C2H2	-	IIe/E	COILS Coil: 275–295
*VvWRKY55*	185	3	0.011	45–101	WRKYGQK	C2H2	-	IIe/E	-
*VvWRKY56*	364	3	0.006	134–193	WRKYGQK	C2HC	FF:9	III/A	-
*VvWRKY57*	151	2	0.004	72–129	WRKYGQK	C2H2	FF:3	IIc/B	-
*VvWRKY58*	618	6	0.005	271–329	WRKYGQK	C2H2	FF:2	IIb/D	COILS Coil: 102–129 LxLxLx motif
*VvWRKY59*	347	3	0.004	275–331	WRKYGQK	C2H2	FF:4	IId/E	Zn-cluster: 226–271 LxxLL motif
*VvWRKY60*	551	6	0.008	300–357	WRKYGQK	C2H2	FF:2	IIb/D	COILS Coil: 114–148 LxLxLx motif
*VvWRKY61*	700	8	0.048	346–402 516–573	WRKYGQK	C2H2	FF:6 FF:1	I/C	Frigida-like: 81–147 LxLxLx motif
*VvWRKY62*	746	5	0.015	318–373 533–590	WRKYGQK	C2H2	FF:1 FF:6	I/C	-

**Table 4 ijms-25-06241-t004:** *WRKY* genes identified in the genome of the *V. vinifera* cultivar Pinot Noir cl. PN40024 in different studies.

WRKY Name	PlantTFDB Wang L [44]/Guo [45]	NCBI		Ensembel	Uniprot	According to Wu [49]/ Zhang and Feng [47]
		Refseq	GenBank			
*VvWRKY1*	GSVIVT01012196001 *VvWRKY*11/*VvWRKY1*	XP_002274549.1 *VvWRKY57*	WJZ80117	VIT_01s0011g00720	F6HF79	*VvWRKY1*/*VvWRKY13-1*, *VvWRKY57-1*
*VvWRKY2*	GSVIVT01020060001 *VvWRKY*19/*VvWRKY2*	XP_010652374.1 *VvWRKY72*	WJZ81033	VIT_01s0026g01730	D7TND6	*VvWRKY2*/*VvWRKY72-3*
*VvWRKY3*	GSVIVT01001332001 *VvWRKY*3/*VvWRKY4*	NP_001268110.1 *VvWRKY2*	WJZ81270	VIT_01s0011g00220	F6HYH9	*VvWRKY58*/*VvWRKY2-3*, *VvWRKY3-1*
*VvWRKY4*	GSVIVT01010525001 *VvWRKY*8/*VvWRKY3*	XP_002275576.1 *VvWRKY75*	WJZ91720	VIT_01s0010g03930	D7TB08	*VvWRKY3*/*VvWRKY57-2*
*VvWRKY5*	GSVIVT01019419001 *VvWRKY*17/*VvWRKY5*	XP_010658402.1 *VvWRKY22*	WJZ81903	VIT_02s0025g00420	F6HUN4	*VvWRKY4*/*VvWRKY22-3*
*VvWRKY6*	GSVIVT01019511001 *VvWRKY*18/*VvWRKY6*	XP_002272720.1 *VvWRKY41*	WJZ81983	VIT_02s0025g01280	D7TVE1	*VvWRKY5*/*VvWRKY41*
*VvWRKY7*	GSVIVT01001286001 *VvWRKY2*/*VvWRKY7*	XP_059589595.1 *VvWRKY21*	WJZ82421	VIT_02s0154g00210	D7TN24	*VvWRKY60*/*-*
*VvWRKY8*	GSVIVT01035426001 *VvWRKY*53/*VvWRKY8*	XP_002279407.1 *VvWRKY50*	-	VIT_04s0008g01470	D7STT5	*VvWRKY6*/*VvWRKY50*
*VvWRKY9*	GSVIVT01035884001 *VvWRKY*54/*VvWRKY9*	XP_010648680.1 *VvWRKY18*	WJZ85291	VIT_04s0008g05750	F6H3I5	*VvWRKY7*/*VvWRKY18*
*VvWRKY10*	GSVIVT01035885001 *VvWRKY*55/*VvWRKY10*	XP_010648274.1 *VvWRKY40*	WJZ85292	VIT_04s0008g05760	F6H3I6	*VvWRKY8*/*VvWRKY40-2*
*VvWRKY11*	GSVIVT01035965001 *VvWRKY*56/*VvWRKY* 11	XP_010648749.1 *VvWRKY32*	WJZ85365	VIT_04s0008g06600	F6H336	*VvWRKY9*/*VvWRKY32-2*
*VvWRKY12*	GSVIVT01033188001 *VvWRKY*48/*VvWRKY12*	XP_002262775.1 *VvWRKY17*	WJZ85515	VIT_04s0069g00920	A0A1U8AHW6	*VvWRKY10*/*VvWRKY11-1*
*VvWRKY13*	GSVIVT01033194001 *VvWRKY*49/*VvWRKY13*	XP_002263836.1 *VvWRKY51*	WJZ85520	VIT_04s0069g00970	D7T0E7	*VvWRKY11*/*VvWRKY51-3*
*VvWRKY14*	GSVIVT01033195001 *VvWRKY*50/*VvWRKY14*	XP_003631843.1 *VvWRKY43*	WJZ85521	VIT_04s0069g00980	A0A438DL90 D7T0E8	*VvWRKY12*/*VvWRKY51-1*
*VvWRKY15*	GSVIVT01019109001 *VvWRKY*16/*VvWRKY15*	XP_059592356.1 *VvWRKYsusiba2*	WJZ85872	VIT_04s0023g00470	F6GX25	*VvWRKY13*/*VvWRKY2-1*
*VvWRKY16*	GSVIVT01034968001 *VvWRKY*52/*VvWRKY16*	XP_002279385.1 *VvWRKY48*	WJZ86707	VIT_05s0077g00730	A0A438GHD0 D7SYJ2	*VvWRKY14*/*VvWRKY48*
*VvWRKY17*	GSVIVT01025491001 *VvWRKY*29/*VvWRKY17*	XP_003632174.3 *VvWRKY3*	WJZ88411	VIT_06s0004g00230	F6GUN9	*VvWRKY15*/*VvWRKY2-4*
*VvWRKY18*	GSVIVT01024624001 *VvWRKY*28/*VvWRKY18*	XP_002272040.1 *VvWRKY24*	WJZ89183	VIT_06s0004g07500	F6GUH8	*VvWRKY16*/*VvWRKY33-2*
*VvWRKY19*	GSVIVT01000752001 *VvWRKY*1/*VvWRKY19*	XP_002282258.1 *VvWRKY21*	WJZ90025	VIT_07s0141g00680	F6GXM5	*VvWRKY17*/*VvWRKY21*
*VvWRKY20*	GSVIVT01028129001 *VvWRKY*34/*VvWRKY20*	XP_002270750.3 *VvWRKY65*	WJZ90472	VIT_07s0005g01520	D7U2J0	*VvWRKY18*/*VvWRKY22-1*
*VvWRKY21*	GSVIVT01028147001 *VvWRKY*35/*VvWRKY21*	XP_002277882.1 *VvWRKY23*	WJZ90489	VIT_07s0005g01710	D7U2K4	*VvWRKY19*/*VvWRKY23*
*VvWRKY22*	GSVIVT01028244001 *VvWRKY*36/*VvWRKY22*	XP_002281194.1 *VvWRKY47*	WJZ90578	VIT_07s0005g02570	F6HZF7	*VvWRKY20*/*VvWRKY47*
*VvWRKY23*	GSVIVT01022067001 *VvWRKY*24/*VvWRKY23*	XP_002283219.1 *VvWRKY7*	WJZ91947	VIT_07s0031g00080	F6H4G0	*VvWRKY21*/*VvWRKY7-2*
*VvWRKY24*	GSVIVT01022245001 *VvWRKY*25/*VvWRKY24*	XP_010652864.1 *VvWRKY51*	WJZ92108	VIT_07s0031g01710	F6H4B4	*VvWRKY22*/*VvWRKY51-2*, *VvWRKY51-4*
*VvWRKY25*	GSVIVT01022259001 *VvWRKY*26/*VvWRKY25*	XP_002279024.1 *VvWRKY13*	WJZ92120	VIT_07s0031g01840	A0A438KK47 D7SW85, I0AVQ1	*VvWRKY23*
*VvWRKY26*	-/*-*	-	WJZ92812	-	-	-/*-*
*VvWRKY27*	GSVIVT01030258001 *VvWRKY*43/*VvWRKY26*	XP_019077410.1 *VvWRKY26*	WJZ92895	VIT_08s0058g00690	F6GXS4	*VvWRKY24*/*VvWRKY33-1*
*VvWRKY28*	GSVIVT01030174001 *VvWRKY*42/*VvWRKY27*	XP_002272504.1 *VvWRKY70*	WJZ92963	VIT_08s0058g01390	F6GXW4	*VvWRKY25*/*VvWRKY70-2*
*VvWRKY29*	GSVIVT01025562001 *VvWRKY*30/*VvWRKY28*	XP_002275978.1 *VvWRKY44*	WJZ93250	VIT_08s0040g03070	F6HQV7	*VvWRKY26*/*VvWRKY44*
*VvWRKY30*	GSVIVT01034148001 *VvWRKY*51/*VvWRKY29*	XP_002270859.1 *VvWRKY49*	WJZ93336	VIT_08s0007g00570	A0A438HSE3 D7THM0	*VvWRKY27*/*VvWRKY49*
*VvWRKY31*	GSVIVT01015952001 *VvWRKY*14/*VvWRKY30*	NP_001267919.1 *VvWRKY40*	WJZ95373	VIT_09s0018g00240	F6HBV8	*VvWRKY28*/*VvWRKY4*, *VvWRKY40-1*
*VvWRKY32*	GSVIVT01012682001 *VvWRKY*12/*VvWRKY31*	XP_002263115.1 *VvWRKY31*	WJZ95843	VIT_10s0116g01200	F6H7H0	*VvWRKY29*/*VvWRKY6-2*
*VvWRKY33*	GSVIVT01007006001 *VvWRKY4*/*VvWRKY59*	XP_010647039.2 *VvWRKY20*	WJZ96075	VIT_00s0463g00010	A0A438DDY6	*VvWRKY59*/*VvWRKY20-1*, *VvWRKY20-4*
*VvWRKY34*	GSVIVT01021252001 *VvWRKY*21/*VvWRKY32*	XP_002269267.1 *VvWRKY65*	WJZ96565	VIT_10s0003g01600	D7TJP4	*VvWRKY30*/*VvWRKY65-1*
*VvWRKY35*	GSVIVT01021397001 *VvWRKY*22/*VvWRKY33*	XP_002272089.1 *VvWRKY* 71	WJZ96695	VIT_10s0003g02810	A5BVH3 D7TK05	*VvWRKY31*/*VvWRKY28-1*
*VvWRKY36*	GSVIVT01021765001 *VvWRKY*23/*VvWRKY34*	XP_002269170.1 *VvWRKY14*	WJZ97031	VIT_10s0003g05740	A0A438BWU0	*VvWRKY32*/*VvWRKY14*
*VvWRKY37*	GSVIVT01023600001 *VvWRKY*27/*VvWRKY35*	XP_002276194.1 *VvWRKY32*	WJZ98338	VIT_11s0037g00150	D7U1A8	*VvWRKY33*/*VvWRKY32-1*
*VvWRKY38*	GSVIVT01029265001 *VvWRKY*39/*VvWRKY36*	XP_002266188.1 *VvWRKY51*	WJZ98778	VIT_11s0052g00450	D0V9L3	*VvWRKY34*/*VvWRKY11-2, VvWRKY11-3*
*VvWRKY39*	GSVIVT01020864001 *VvWRKY*20/*VvWRKY37*	XP_002283603.1 *VvWRKY71*	WJZ98954	VIT_12s0028g00270	E0CU42	*VvWRKY35*/*VvWRKY28-2*
*VvWRKY40*	-/*-*	XP_002277383.1 *VvWRKY65*	WJZ99058	-	-	*VvWRKY36*/*VvWRKY3-2, VvWRKY65-2, VvWRKY65-3*
*VvWRKY41*	GSVIVT01030453001 *VvWRKY*44/*VvWRKY38*	XP_002269696.2 *VvWRKY* 31	WJZ99341	VIT_12s0059g00880	F6HIC7	*VvWRKY37*/*VvWRKY6-1*
*VvWRKY42*	GSVIVT01030046001 *VvWRKY*41/*VvWRKY39*	XP_002272407.1 *VvWRKY20*	WJZ99676	VIT_12s0057g00550	F6HHL8	*VvWRKY38*/*VvWRKY20-3, VvWRKY20-6*
*VvWRKY43*	GSVIVT01029688001 *VvWRKY*40/*VvWRKY40*	XP_010657556.1 *VvWRKY9*	WKA00085	VIT_12s0055g00340	F6H1R3	*VvWRKY39*/*VvWRKY9*
*VvWRKY44*	GSVIVT01032662001 *VvWRKY*46/*VvWRKY41*	XP_002275373.1 *VvWRKY55*	WKA00784	VIT_13s0067g03130	A0A438E3U8 F6HC34	*VvWRKY40*/*VvWRKY55*
*VvWRKY45*	GSVIVT01032661001 *VvWRKY*45/*VvWRKY42*	XP_002275401.1 *VvWRKY70*	WKA00785	VIT_13s0067g03140	F6HC33	*VvWRKY41*/*VvWRKY70-1*
*VvWRKY46*	GSVIVT01036223001 *VvWRKY*57/*VvWRKY43*	XP_002270614.2 *VvWRKY74*	WKA03253	VIT_14s0081g00560	F6HVI7	*VvWRKY42*/*VvWRKY74*
*VvWRKY47*	GSVIVT01033063001 *VvWRKY*47/*VvWRKY44*	XP_002274387.1 *VvWRKY75*	WKA04134	VIT_14s0068g01770	D7SVN0 I3RQB5	*VvWRKY43*/*VvWRKY45*
*VvWRKY48*	GSVIVT01011356001 *VvWRKY*9/*VvWRKY45*	XP_002277221.2 *VvWRKY72*	WKA04564	VIT_14s0108g00120	D7SX70	*VvWRKY44*/*VvWRKY72-2*
*VvWRKY49*	GSVIVT01011472001 *VvWRKY*10/*VvWRKY46*	XP_010661104.2 *VvWRKY4*	WKA04672	VIT_14s0108g01280	-	*VvWRKY45*/*-*
*VvWRKY50*	GSVIVT01018300001 *VvWRKY*15/*VvWRKY47*	XP_002270527.1 *VvWRKY12*	WKA05315	VIT_15s0021g01310	A0A1U6ZIF2	*VvWRKY46*/*VvWRKY12-1, VvWRKY12-2*
*VvWRKY51*	GSVIVT01027069001 *VvWRKY*33/*VvWRKY48*	XP_002281031.1 *VvWRKY46*	WKA05852	VIT_15s0046g01140	F6I6B1	*VvWRKY47*/*VvWRKY46*
*VvWRKY52*	GSVIVT01026969001 *VvWRKY*32/*VvWRKY49*	XP_002275528.3 *VvWRKY24*	WKA05934	VIT_15s0046g02150	D7UCE6 A0A438JLT2	*VvWRKY48*/*VvWRKY24*
*VvWRKY53*	GSVIVT01026965001 *VvWRKY*31/*VvWRKY50*	XP_002276925.1 *VvWRKY22*	WKA05938	VIT_15s0046g02190	D7UCE2 A0A438JM47	*VvWRKY49*/*VvWRKY22-2*
*VvWRKY54*	GSVIVT01028823001 *VvWRKY*38/*VvWRKY51*	XP_010662789.1 *VvWRKY22*	WKA07335	VIT_16s0050g01480	E0CUS7	*VvWRKY50*/*-*
*VvWRKY55*	-/*-*	XP_010662788 *VvWRKY27*	WKA07336	-	-	-/*-*
*VvWRKY56*	GSVIVT01028718001 *VvWRKY*37/*VvWRKY52*	XP_002267793.2 *VvWRKY53*	WKA07458	VIT_16s0050g02510	E0CUJ8	*VvWRKY51*/*VvWRKY53*
*VvWRKY57*	GSVIVT01008553001 *VvWRKY*6/*VvWRKY53*	NP_001268218.1 *VvWRKY1*	WKA07893	VIT_17s0000g01280	Q5IZC7	*VvWRKY52*/*VvWRKY1-2*, *VvWRKY75*
*VvWRKY58*	GSVIVT01008046001 *VvWRKY5*/*VvWRKY54*	XP_010663394.1 *VvWRKY72*	WKA08379	VIT_17s0000g05810	D7SIE7	*VvWRKY53*/*VvWRKY72-1*
*VvWRKY59*	GSVIVT01009441001 *VvWRKY7*/*VvWRKY55*	XP_002284966.1 *VvWRKY7*	WKA09909	VIT_18s0001g10030	E0CPR7	*VvWRKY54*/*VvWRKY7-1*
*VvWRKY60*	GSVIVT01037686001 *VvWRKY58*/*VvWRKY56*	XP_010644476.1 *VvWRKY31*	WKA12500	VIT_19s0090g00840	F6HEQ5	*VvWRKY55*/*VvWRKY42*
*VvWRKY61*	GSVIVT01037775001 *VvWRKY59*/*VvWRKY57*	XP_010644520.1 *VvWRKY20*	WKA12584	VIT_19s0090g01720	F6HER4	*VvWRKY56*/*VvWRKY20-2*, *VvWRKY20-5*
*VvWRKY62*	GSVIVT01014854001 *VvWRKY13*/*VvWRKY58*	XP_002265612.1 *VvWRKY2*	-	VIT_19s0015g01870	F6I4X4	*VvWRKY57*/*VvWRKY2-2*

## Data Availability

Publicly available datasets were analyzed in this study. This data can be found here: https://www.ncbi.nlm.nih.gov (accessed on 22 February 2024). The accession numbers of genome data are represented in the article.

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
