# Peer review of "Genome-Wide Identification, Characterisation, and Evolution of the Transcription Factor WRKY in Grapevine (Vitis vinifera): New View and Update"

_ijms, 2024, doi:10.3390/ijms25116241_

Round 1

Reviewer 1 Report

Comments and Suggestions for Authors

 Although the study carried out a systematic bioinformatics analysis of the WRKY family of grapes, what are the specific expression characteristics of these members at different stages of grape development? How are they related to evolution?what is the significance of the differences in the varieties.  This manuscript could insufficiently support the conclusions to be published in this TOP scientific journal.

Author Response

We thank the Reviewer very much for the careful reading of our manuscript and for providing useful comments. Our comments are given below.

REV: Although the study carried out a systematic bioinformatics analysis of the WRKY family of grapes, what are the specific expression characteristics of these members at different stages of grape development? How are they related to evolution? what is the significance of the differences in the varieties.

Author: We fully agree with the Reviewer's comments. Indeed, it is very interesting to see the correlation between the identified characteristics of different phylogroups and expression profiles. This investigation is in our immediate plans. In addition, we‘re going to analyse the grape pan-transcriptome and see the effect of inter-varietal nucleotide variability on expression evaluation in different varieties.

This study focuses only on the identification of WRKY genes and their evolution. This is because confusion with the naming of genes appears in new works, which leads to more difficulties later on. For example, in work on transgenic grapes, it was shown that the WRKY70 (VIT_13s0067g03140) is involved in norisoprenoid and flavonol biosynthesis [Wei et al, 2023]. At the same time, the name WRKY70 is found in the RefSeq database under two IDs: XP_00227272504 (in our study VvWRKY28) and XP_002275401 (in our study VvWRKY45). Thus, naming VvWWRKY70 could lead to errors in further studies. The situation with the VvWRKY35 gene is different. Its role in tolerance to cold and salt stress in transgenic Arabidopsis was previously studied, where it was described as VvWRKY28 [Liu et al., 2022]. In another paper investigating the functions of WRKYs, this gene (GSVIVT01021397001) is referred to as VvWRKY22 [Zhu 2016], and in a paper investigating WRKYs in strawberries, it is included in the phylogenetic analysis as VvWRKY71 (XP_002272089) [Yue 2022]. At the same time, another VvWRKY71 with accession number XP_002283603 is deposited in the RefSeq database and analysed under this name in another paper on grape [Hou 2013]. A recent paper was published showing that VvWRKY71 could promote the biosynthesis of proanthocyanidins, but it is extremely difficult to understand which RefSeq ID was used to annotate the assembled transcriptome [Feng et al., 2024]. Such situations show the extreme necessity of using a unified WRKY gene numbering.

In addition, new assemblies of grape genomes provide the opportunity to make a more comprehensive study, and improving bioinformatics methods have allowed the identification of new chimeric genes, which may be useful in planning further studies.

Reviewer 2 Report

Comments and Suggestions for Authors

In the present study, the newly uploaded genomic assemblies databases for different grapevine cultivars was use to identifying the WRKY genes, and then their bioinformatics characteristics were also analyzed. The study was well-designed and the manuscript was well-written. Some suggests were put forwarded as follows:

In Title, I suggest using update, but not revision.

In the section of Abstract, the authors claimed the experiments, and the results should be more described and concluded.

In the section of Introduction, Line 67-84, the paragraphs should be reorganized to be clearer.

Line 306-307, please compared the current data with the reports.

Line 310, delete “All data are presented in Table 4”, instead, describe the table.

Lien 355-365, the statements are confused. For example but not limited, Line 355, the present study or previous reports? Line 359, the authors of this study or previous studies?

Line 545, Twenty-three.

The description of the Material and Methods should be more detailed.

Author Response

We thank the Reviewer very much for the careful reading of our manuscript and for providing useful comments. We revised the manuscript in accordance with the comments of the Reviewer. Our comments and answers are given below.

REV: In Title, I suggest using update, but not revision.

Author: We agree with this proposal.

REV: In the section of Abstract, the authors claimed the experiments, and the results should be more described and concluded.

Author: We changed Abstract according to the Reviewer suggestion.

REV: In the section of Introduction, Line 67-84, the paragraphs should be reorganized to be clearer.

Author: We rewrote text according to the Reviewer suggestion

REV: Line 306-307, please compared the current data with the reports.

Author: We have modified this paragraph and added descriptions of the data. Since the Table 4 contains the names of genes and the column name indicates according to which author, we have noted the references in the Table 4 itself.

REV: Line 310, delete “All data are presented in Table 4”, instead, describe the table.

Author: We added the description of the Table 4 to the text.

REV: Lien 355-365, the statements are confused. For example but not limited, Line 355, the present study or previous reports? Line 359, the authors of this study or previous studies?

Author: These assumptions have been made in other works, and they show the need to investigate inter-varietal nucleotide variability. This paragraph has been corrected more accurately.

REV: Line 545, Twenty-three.

Author: We did.

REV: The description of the Material and Methods should be more detailed.

Author: We added some information to the Material and Methods

Reviewer 3 Report

Comments and Suggestions for Authors

The manuscript by Vodiasova et al. focus on the WRKY members in Vitis vinifera, and emphasize their gene structure and amino acid variability between different grape cultivars. Additionally, the authors explore the evolution of the grape WRKYs based on the phylogenetic analysis. These results provide scientific information of WRKY members in grape for readers.

Although this paper is well written, several questions should be solved clearly, and the authors should be more considerate about their conclusions and claims according to their data.

Major comments:

In the abstract, the authors emphasize their founds were the first. In my knowledge, the opinions have been proposed previously, especially L19-20. Please check it and be more objective about your claims.

The authors numbered the WRKYs in grape and renamed them. Please clarify how the WRKYs are numbered, based on the position on the chromosomes? In addition, the names of WRKYs whose function were verified in previous studies should be remained. For example, the WRKY70 (VIT_13s0067g03140) has been proved to be functional in norisoprenoid and flavonol biosynthesis by Wei et al. (Transcription factor VvWRKY70 inhibits both norisoprenoid and flavonol biosynthesis in grape. Plant Physiol. 2023 193:2055-2070.). Name of this gene should be WRKY70 other than WRKY45.

In L138, how the number 965 comes from and what it means? Please clarify it.

Minor comments:

In this manuscript, several references are cited by mistake. For example, in Table 4 the cite should be Wu et al., 2022. Please check the cites in the whole files.

Please add the explanation of the abbreviations. The FF in the Table and Figure might mean Functional Family?

Comments on the Quality of English Language

It is ok.

Author Response

We thank the Reviewer very much for the careful reading of our manuscript and for providing useful comments. We revised the manuscript in accordance with the comments of the Reviewer. Our comments and answers are given below.

REV: In the abstract, the authors emphasize their founds were the first. In my knowledge, the opinions have been proposed previously, especially L19-20. Please check it and be more objective about your claims.

Author: Thank you to the Reviewer for this comment. We’ve rewritten the Abstract and the part about chimeric WRKYs in Discussion. We've clarified that the two chimeric WRKY genes were first found (with N-terminal signal peptide region followed by a non-cytoplasmic domain and with Frigida-like domain). The amino acid variability between different grape cultivars hasn't been studied before.

REV: The authors numbered the WRKYs in grape and renamed them. Please clarify how the WRKYs are numbered, based on the position on the chromosomes? In addition, the names of WRKYs whose function were verified in previous studies should be remained. For example, the WRKY70 (VIT_13s0067g03140) has been proved to be functional in norisoprenoid and flavonol biosynthesis by Wei et al. (Transcription factor VvWRKY70 inhibits both norisoprenoid and flavonol biosynthesis in grape. Plant Physiol. 2023 193:2055-2070.). Name of this gene should be WRKY70 other than WRKY45.

Author: For each gene encoding the corresponding WRKY protein, its position on the chromosome was determined according to each grape cultivar genome.  This information is in Supplementary 2. WRKY TFs were numbered according to the ordinal chromosome location number (in most of the assemblies analysed). We added this explanation to the manuscript.

Thanks to the reviewer for the above article, which we missed. We have added this paper and reviewed other studies on the functions of the WRKY transcription factor in the Introduction. The paragraph in the Discussion about the need for uniform numbering was also expanded.  For example, the mentioned gene VvWRKY70 is found in the RefSeq database under two IDs: XP_00227272504 (in our study VvWRKY28) and XP_002275401 (in our study VvWRKY45). The situation is even more complicated with the VvWRKY35 gene. Its role in tolerance to cold and salt stress in transgenic Arabidopsis was previously studied, where it was described as VvWRKY28 (Liu et al., 2022). However, in the RefSeq database it is VvWRKY71 (XP_002272089), although there is another VvWRKY71 (XP_002283603) in the same database and they are not homologues. What complicates the situation with the numbering is the fact that recently a paper was published showing that VvWRKY71 could promote the biosynthesis of proanthocyanidins (Feng et al., 2024). And this further demonstrates the need for uniform numbering. At the same time, to avoid difficulties with previous numbering, Table 2 lists all possible IDs and WRKY gene numbers from other papers as completely as possible.

REV: In L138, how the number 965 comes from and what it means? Please clarify it.

Author: We explained it in the text.

REV: In this manuscript, several references are cited by mistake. For example, in Table 4 the cite should be Wu et al., 2022. Please check the cites in the whole files.

Author: We changed it and checked all files.

REV: Please add the explanation of the abbreviations. The FF in the Table and Figure might mean Functional Family?

Author: We added the explanation of the abbreviations in the text.

Reviewer 4 Report

Comments and Suggestions for Authors

This is a paper describing identification, classification, and characterization of the transcription factor WRKY of grapevine. Previous studies on identification and classification of WRKY of grapevine were performed using a single cultive (Pinot Noir). In the current study, the authors have used multiple grapevine cultivars and provided new findings on the WRKY family in grapevine. The manuscript is well-written. There are only minor concerns that should be addressed.

Figure 1(a): I suggest that the authors add the term wound to Mechanical injury.

Gene names of subgroup D1, D2, and D3 are different between Figure 2, Figure 3, and Supplementary 2. 

Comments on the Quality of English Language

There are minor typographical errors throughout the manuscript. 

Author Response

Reviewer 4

We thank the Reviewer very much for the careful reading of our manuscript and for providing useful comments. We revised the manuscript in accordance with the comments of the Reviewer. Our comments and answers are given below.

REV: Figure 1(a): I suggest that the authors add the term wound to Mechanical injury.

Author: We’ve added to the Figure 1.

REV: Gene names of subgroup D1, D2, and D3 are different between Figure 2, Figure 3, and Supplementary 2.

Author: We've corrected the gene names of  subgroup D1, D2, and D3 in Supplementary 2 according to the  Figure 2, Figure 4 and text.

Round 2

Reviewer 1 Report

Comments and Suggestions for Authors

Thank you for your answers and provided detail experiments. I think this paper can be accepted.

Reviewer 2 Report

Comments and Suggestions for Authors

The authors have well-addressed each comment, and I do no have more revision suggestion.